

# Distributed wind measurements with multiple quadrotor UAVs in the atmospheric boundary layer

Tamino Wetz[1], Norman Wildmann[1], and Frank Beyrich[2]

[1]Deutsches Zentrum für Luft- und Raumfahrt e.V., Institut für Physik der Atmosphäre, Oberpfaffenhofen, Germany
[2]Deutscher Wetterdienst, Meteorologisches Observatorium Lindenberg – Richard-Aßmann-Observatorium, Lindenberg, Germany

**Correspondence:** Tamino Wetz (tamino.wetz@dlr.de)

**Abstract.** In this study, a swarm of quadrotor UAVs is presented as a system to measure the spatial distribution of atmospheric boundary layer flow. The big advantage of this approach is, that multiple and flexible measurement points in space can be sampled synchronously. The algorithm to obtain horizontal wind speed and direction is designed for hovering flight phases and is based on the principle of aerodynamic drag and the related quadrotor dynamics. During the FESST@MOL campaign at the

Boundary Layer Field Site (Grenzschichtmessfeld, GM) Falkenberg of the Lindenberg Meteorological Observatory - Richard-Aßmann-Observatory (MOL-RAO), 76 calibration and validation flights were performed. The 99 m tower equipped with cup and sonic anemometers at the site is used as the reference for the calibration of the wind measurements. The validation with an independent dataset against the tower anemometers reveals that an average accuracy of $\sigma_{rms} < 0.3\,\mathrm{m\,s^{-1}}$ for the wind speed and $\sigma_{\mathrm{rms},\psi} < 8°$ for the wind direction was achieved. Furthermore, we compare the spatial distribution of wind measurements

with the swarm to the tower vertical profiles and Doppler wind lidar scans. We show that the observed shear in the vertical profiles matches well with the tower and the fluctuations on short time scales agree between the systems. Flow structures that appear in the time series of a line-of-sight measurement and a two-dimensional vertical scan of the lidar can be observed with the swarm and are even sampled with a higher resolution than the deployed lidar can provide.

## 1   Introduction

Wind patterns and flow structures in the atmospheric boundary layer (ABL) are diverse and complex, depending on the synoptic conditions, mesoscale forcings and local effects (e.g. changes in land use or topographic changes). Examples for such flow structures are convective elements (Kaimal et al., 1976), coherent structures due to canopy flows (Dupont and Brunet, 2009), recirculation zones in mountainous terrain (Menke et al., 2019) or even a mix of convective and terrain-driven flows (Brötz et al., 2014). Turbulent structures also occur in the interaction of the ABL with wind turbines. Following the review of Veers

et al. (2019) one of the major challenges in the science of wind energy is the understanding of the microscale wind conditions around a wind plant. This means the inflow conditions as well as the complex wake pattern behind individual turbines and their interaction in wind parks. The goal of the project presented in this study is to provide a tool for flexible measurements in this field.



A variety of measurement campaigns were performed in the past, studying the wind around wind plants using different

measurement techniques (Rajewski et al., 2013; Fernando et al., 2019; Wilczak et al., 2019) including meteorological masts, lidar (Wildmann et al., 2018), or airborne in-situ measurements (Platis et al., 2018). These methods provide valuable results, but are associated with a significant logistical effort and are not very flexible in their deployment. Against these drawbacks, unmanned aerial vehicles (UAV) are getting more relevant in supporting or expanding conventional atmospheric measurement techniques. The flexibility in flight patterns is almost unlimited. Furthermore, by applying multiple UAVs simultaneously at a

campaign, there is the potential of measuring atmospheric quantities simultaneously and in-situ at flexible positions that were not possible before.

In general, there are two different types of UAV, one with fixed-wing configuration and one that uses only the power of the rotor to provide the lift for flying the vehicle (known as rotary-wing UAVs). Both types of UAVs were already applied for measuring the wind speed in the lower atmosphere (see for example Wildmann et al. (2015) for fixed-wing UAVs, Cuxart et al.

(2019) for rotary-wing or Kral et al. (2020) for a combination of both). The purpose of the present project is to measure the wind simultaneously at different positions in predefined patterns. For this purpose, UAV rotary-wing systems are chosen over those with fixed wings. Multirotors are able to hover at fixed positions and need only small space for take-off and landing.

For measuring both, wind speed and direction using rotary-wing UAVs, different methods have been described in literature. There are two major concepts: The first approach measures wind with an additional external wind sensor e.g. sonic anemome-

ters (Shimura et al., 2018; Reuter et al., 2020; Thielicke et al., 2020; Nolan et al., 2018) or hot wire/element probes (Cuxart et al., 2019; Molter and Cheng, 2020). The second approach uses only on-board sensors of the avionic system of a multirotor, e.g. the orientation angles measured by an inertial measurement unit (IMU) are used for wind measurement (Palomaki et al., 2017; Brosy et al., 2017; Neumann et al., 2012; Neumann and Bartholmai, 2015; Gonzalez-Rocha et al., 2017; González-Rocha et al., 2019; Wang et al., 2018; Bartholmai and Neumann, 2011; Bell et al., 2020). Furthermore there are commercial

solutions for measuring the wind of the lower atmosphere with multirotors (Greene et al., 2019). A comparison of UAVs used in atmospheric science is exercised by Barbieri et al. (2019). Abichandani et al. (2020) compare different approaches described in the literature and demonstrate that with their best approach for using only on-board sensors a root mean square deviation of $\epsilon_{\mathrm{rms}} = 0.6 \ \mathrm{m \, s^{-1}}$ in wind speed is determined.

Regarding the method for measuring the wind with multirotors using only on-board sensors Neumann and Bartholmai

(2015) tried to link the wind-speed with the inclination angle of the multirotor by taking the well-known Rayleigh-drag-equation into account. They tried to estimate the unknown drag coefficient and projected area of the multirotor by wind-tunnel tests and analytical approaches. The wind-tunnel tests were performed with still rotors. They concluded that neglecting the rotor movement is not a valid approach for estimating the drag coefficient. This is confirmed by wind-tunnel tests of Schiano et al. (2014). In their experiment they investigated the drag coefficient for different yaw and pitch angles of the multirotor.

However, the experiments were performed with still rotor which had a significant influence on the results, compared to real flight environments with moving rotors. Therefore, Neumann and Bartholmai (2015) calibrate the wind-speed directly against the inclination angle without estimating a drag coefficient and came up with a polynomial fit of second order. Brosy et al. (2017) use the GPS velocity as reference speed for obtaining a regression function between wind speed and inclination angle.





They performed flights with different constant velocities in calm wind conditions. The obtained relation is a root-function which is only valid to the limit of $6\,\mathrm{m\,s^{-1}}$. Further, González-Rocha et al. (2019) claims a linear relation between wind-speed and inclination angle for their multirotor as demonstrated by wind-tunnel experiments.

In the present study we introduce a method to derive the wind using a similar approach, which we describe in detail in Sect. 4. The hardware and software of the quadrotors is introduced in Sect. 2 and the experiment in which ten UAVs were operated simultaneously is described in Sect. 3. Both, wind speed and direction are calibrated against sonic anemometers for the ten quadrotors and the accuracies of different calibration datasets are validated with independent validation datasets (Sect. 5). Measured wind data from the swarm of multirotors is compared to cup and sonic anemometer measurements as well as Doppler wind lidar measurements to evaluate the capabilities to resolve microscale structures in the ABL (Sect. 6).

## 2 System description

This section describes the measurement system including the hardware and necessary software for performing simultaneous wind field measurements with rotary wing UAVs.

### 2.1 UAV hardware

In general, commercial UAVs have some essential sensors implemented for their avionics. These are at least an IMU, i.e. gyroscopes, accelerometers and magnetometers to measure the attitude, as well as a GNSS system to determine the position. The flight controller (or autopilot) processes the measured data for either stabilizing the UAV to hold the position in hover state (together with the data from the GNSS) or flying pre-defined trajectories. Actuator outputs of the autopilot are the rotor speeds.

For our purpose of wind field measurements with a swarm of UAVs in the ABL, we chose the "Holybro QAV250" quadrotor frame in combination with the Pixhawk®4 Mini flight controller as shown in Fig. 1. This system has multiple advantages and meets most of the requirements that were defined:

- All raw output data of the IMU should be accessible which requires an open-source solution such as the Pixhawk flight controller and the PX4 software. With the PX4 software, the raw sensor data are available at 100 Hz. For wind measurement, we average the data to 1 Hz in this study. The selection of an open-source solution has the further advantage that the system allows software adjustment and the possibility of implementing additional sensors.

- The system should be as simple as possible regarding the flight kinematics, for calculating and calibrating wind speed measurement. Thus, a suitable type is a multirotor consisting of only four rotors, i.e. a quadrotor.

- The UAV should endure strong winds with a good stability of the hover position. For this reason we selected racing drones which are designed to fly with high velocities while still having a high level of agility. This type has the potential to react fast against small disturbances and we thus expect it to be able to resolve small scales of the flow. Since it is desirable to resolve structures as small as possible, the small class of racing drones with a distance between the rotor axes of 0.25 m was chosen.





– A long flight duration is desirable to capture all relevant turbulent scales in the ABL. A typical averaging period for turbulence retrievals is 30-minutes. This flight time can not be reached with the current combination of battery, airframe and controller parameters, but the 12-minutes that are currently possible with a battery capacity of 2.600 mAh can likely be increased with optimization of hard- and software in future.

   – Taking the goal of a swarm of UAVs into account, the single quadrotor should be off-the-shelf.

– With a weight of $m = 0.65$ kg, the quadrotor is below the weight of 2 kg which defines the threshold in Germany above which special permits and a piloting license are needed to operate UAVs even below 100 m and in safe airspaces.

In addition to the on-board sensors of the flight controller, a temperature and humidity sensor of type HYT271 is integrated in every quadrotor. Swarm communication is realized by a Wifi router to which all quadrotors and the ground station computer are connected. The important system parameters are listed in Table 1.

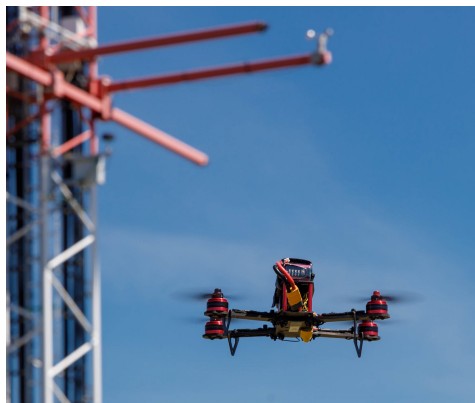

**Figure 1.** Picture of quadrotor "HolyBro QAV250" in front of the mast at Falkenberg. Photo by Bernd Lammel.

**Table 1.** System description quadrotor "Holybro QAV250".

| parameter | description |
|---|---|
| UAV-type | racing-quadrotor |
| weight (incl. battery) | 0.65 kg |
| dimension (axis to axis) | 0.25 m |
| autopilot | Pixhawk 4 Mini |
| temp. and humidity-sensor | HYT271 |
| frequency of sensor-data-logging | up to 100 Hz |
| mission flight speed | $<15 \, \mathrm{m \, s^{-1}}$ |
| flight time | <12 min |



We will refer to the swarm of Holybro QAV250 quadrotors as the SWUF-3D (Simultaneous Wind measurement with Unmanned Flight Systems in 3D swarms) corresponding to the name of the project.

## 2.2 Software

The Pixhawk 4 Mini autopilot features the PX4 software. Specific parameter settings for the quadrotor are set to optimize flight behavior and to realize swarm flights. In order to align the quadrotor to the wind direction, the "weather-vane" mode is enabled.

In that mode, the yaw angle is used as a control variable to minimize the roll angle amplitude and hence the quadrotor will always face in upwind direction. The minimum roll angle for weather-vane controller to demand a yaw-rate is set to $1°$.

Control of the swarm is realized by the software © QGroundControl. This ground station software is used to launch and monitor the swarm. The flight paths are planned a priori in global coordinates and are uploaded to the single quadrotors. This allows complete freedom in the design of possible flight patterns. However, it has to be guaranteed by design that flight paths

do not cross and thus no collision of UAVs is possible. All flight data that is logged by the autopilot to the internal SD-card can be transferred to the ground station through an interface in QGroundControl.

## 3 The FESST@MOL experiment

Originally, calibration and validation flights with the SWUF-3D were planned to be performed during the FESSTVaL (Field Ex-

periment on sub-mesoscale spatio-temporal variability in Lindenberg) campaign that was initiated by the Hans-Ertel-Zentrum für Wetterforschung (HErZ) of the German Meteorological Service (Deutscher Wetterdienst, DWD). Due to the Sars-CoV-2 pandemics, this campaign could not be realized as planned, but had to be significantly reduced in the number of participants and divided into smaller sub-campaigns (so called "FESST@home" experiments). From May to August 2020, the campaign FESST@MOL was organized at the Boundary Layer Field Site (Grenzschichtmessfeld, GM) Falkenberg of the Lindenberg

Meteorological Observatory - Richard-Aßmann-Observatory (MOL-RAO). GM Falkenberg is located about 80 km to the South-East of the centre of Berlin. Here, DWD runs a comprehensive operational measurement program of micrometeorological and boundary layer measurements including the use of a variety of wind sensors and measurement systems (cup and sonic anemometers at towers, Doppler sodar, Doppler lidar, see, e.g. Neisser et al. (2002); Beyrich and Adam (2007)). During FESST@MOL, this measurement program was extended by the operation of six Doppler lidar systems provided by DLR and

KIT (Karlsruhe Institute of Technology). It was a major goal of this campaign to test and to compare different scanning configurations to derive both wind and turbulence information from Doppler lidar measurements and to elaborate strategies for the validation of the Doppler lidar retrievals by airborne measurements.

The central measurement facility at GM Falkenberg is a 99 m tower instrumented with sonic and cup anemometers at multiple levels. Cup anemometers (Thies wind transmitter type 4.3303.022.000) are installed at heights of 10 m, 20 m, 40 m,

60 m, 80 m, and 98 m above ground. At each level, anemometers are mounted at the tips of booms pointing towards $11°$, $191°$, and $281°$, respectively. Wind direction is measured with wind vanes (Thies wind direction transmitter type 4.3121.32.000 /



4.3124.30.002 ) at 40 m and 90 m heights. For the final wind data set, the measurements are taken from those sensors which are not situated in the upstream or downstream region of the tower, depending on the actual wind direction. Three-dimensional sonic anemometer-thermometers (Metek USA-1) are mounted on the booms pointing towards South (191°) at the 50 m and

90 m levels, these measurements are distorted through the tower for wind directions between 345° and 50° via North, but this wind direction was not observed during the present flight campaign.

    The quadrotor flights were realized during the period July 21-31, 2020, at GM Falkenberg. In total 76 SWUF-3D flights were performed with an accumulated flight duration of 4800 minutes (counting every minute of individual quadrotor flights). A protocol of all flights and their basic characteristics (flight time, flight pattern, number of quadrotors, mean wind conditions)

is given in Appendix C. In general, the experiments were performed by flying individual predefined flight paths of multiple quadrotors simultaneously to discrete positions. At these discrete positions, the quadrotors were hovering for a certain time before flying back to the take-off location. Different swarm flight patterns were implemented. All of the pattern were targeting the goal to calibrate and validate the wind measurement algorithm of the quadrotors and of whole the swarm. The flight pattern "drone tower" consisted of up to eight quadrotors hovering at the altitudes of the tower wind measurements. The quadrotors

were flown simultaneously at the same horizontal position marked in Fig. 2 but at different altitudes. The horizontal position was defined close to the tower in upwind direction, in order to have free inflow towards the quadrotor and no disturbance from the tower. A safety distance of 20 m to the tower was chosen. For a second flight pattern, one of the Doppler wind lidars was used for inter-comparison with SWUF-3D measurements. The location of the lidar is indicated in Fig. 2. The lidar is a Leosphere Windcube 200S, it was operated at a physical resolution of 50 m with range-height indicator (RHI) scans and in

staring mode. The staring mode at an elevation angle of 7.1° allowed to place all quadrotors within the lidar beam to measure the same flow field continuously ("lidar line"). The pattern "3x3 lidar" spanned an array of 3x3 quadrotors to represent a 2-D field within the RHI plane of the lidar scan. The mesh-width of the SWUF-3D grid in this configuration was 100 m in the horizontal and 40 m in the vertical. Another pattern that appears in the protocol, called "drone line", was not used in the present analysis, since the distance to the 99 m tower is larger then for the drone tower. However this pattern can get relevant in future

data analysis.

## 4    Methods

### 4.1    System equation

The motion of the quadrotor can be described by the fundamental mechanic equation of force and moment equilibrium. For the definition of the motion of the system, the frames of reference need to be introduced.

#### 4.1.1    Coordinate systems

The inertial frame, or also called earth frame, is fixed on the earth with the z-component pointing orthogonally away from the earth surface. The second frame, the body frame, moves with the system and has its origin in the center of gravity of the





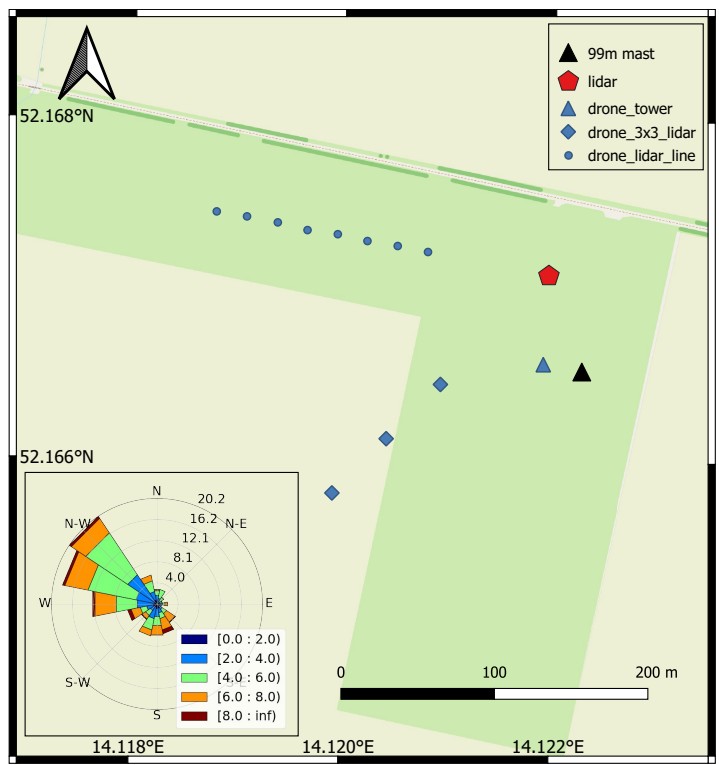

**Figure 2.** Map of the experiment site, including the locations of UAV swarm measurements. ©OpenStreetMap contributors 2020. Distributed under a Creative Commons BY-SA License. The wind rose in the bottom left shows the wind conditions during the campaign period (21-30 July 2020).

quadrotor (see also Palomaki et al., 2017). The inertial frame is distinguished by the indices $\mathbf{i_n}$ with $n(1, 2, 3)$ for the three dimensions, similarly the body frame is defined by the indices $\mathbf{b_n}$. The position vector $\mathbf{X_i}$ and the angular vector $\mathbf{\Phi_i}$ are defined in the inertial frame. Furthermore, $\mathbf{V_b}$ is the vector of translation speeds and $\omega_\mathbf{b}$ the angular velocity vector in the body frame.

$$\mathbf{X_i} = \begin{bmatrix} x & y & z \end{bmatrix}^T \tag{1}$$

$$\mathbf{\Phi_i} = \begin{bmatrix} \phi & \theta & \psi \end{bmatrix}^T \tag{2}$$

$$\mathbf{V_b} = \begin{bmatrix} u & v & w \end{bmatrix}^T \tag{3}$$

$$\omega_\mathbf{b} = \begin{bmatrix} p & q & r \end{bmatrix}^T \tag{4}$$

In order to transform the motions from one frame to another a rotation matrix $\mathbf{R}$ is needed. For detailed definition see Appendix B. The time derivative of the position vector $\dot{\mathbf{X}}_\mathbf{i}$ represents the velocity vector in inertial frame.

$$\dot{\mathbf{X}}_\mathbf{i} = \mathbf{R}(\mathbf{\Phi_i})\mathbf{V_b} \tag{5}$$



### 4.1.2 Mechanical model

Regarding the quadrotor as a rigid body, its motion in space can be described by the basic mechanical equation dividing the motion in translation and rotation. The translation motion is balanced by the gravity force $\mathbf{F_g}$, control forces $\mathbf{F_c}$ and external forces $\mathbf{F_e}$. The inertial forces are defined by the product of mass $m$ and acceleration $\mathbf{\ddot{X}_i}$.

$$m\mathbf{\ddot{X}_i} \quad = \quad \mathbf{F_g} + \mathbf{F_c} + \mathbf{F_e} \tag{6}$$

The angular momentum is driven by control moments $\mathbf{M_c}$ and external moments $\mathbf{M_e}$. Further, the angular inertia $\mathbf{I}$ and the

angular acceleration vector $\mathbf{\ddot{\Phi}_i}$ is needed to define the momentum equation.

$$\mathbf{I\ddot{\Phi}_i} \quad = \quad \mathbf{M_c} + \mathbf{M_e} \tag{7}$$

Transforming the equations of motion in the body frame leads to additional gyroscopic terms ($i_3$ and $b_3$ representing unit vectors in inertial respectively body frame):

$$m\mathbf{\ddot{X}_i} = m\mathbf{\dot{V}_b} + m\mathbf{V_b} \times \omega_{\mathbf{b}} \quad = \quad mg\mathbf{R^T}i_3 - F_c b_3 + \mathbf{F_e} \tag{8}$$

In the following step, only the linear motions were regarded for calculating the wind speed. Further the only external forces $F_e$ are in this case the wind forces $F_w$. For the linear motions in body frame the following equations are obtained:

$$m(\dot{u} + qw - rv) \quad = \quad -mg[\sin(\theta)] + F_{w,x} \tag{9}$$

$$m(\dot{v} + pw - ru) \quad = \quad -mg[\cos(\theta)\sin(\phi)] + F_{w,y} \tag{10}$$

$$m(\dot{w} + pv - qu) \quad = \quad mg[\cos(\theta)\cos(\phi)] + F_{w,z} - d(n_1^2 + n_2^2 + n_3^2 + n_4^2) \quad , \tag{11}$$

where $g$ is the acceleration of gravity, $n_i$ are rotational speeds of the motors and $d$ the thrust coefficient. For our wind estimation in hover state with a weather vane mode that ensures the quadrotor to point into the main wind direction, we proceed with only Eq. (9).

### 4.2 Wind estimation in hover state

The well-known aerodynamic Rayleigh-equation for calculating aerodynamic forces from the wind velocity $V_w$ takes the

projected front area $A$ and the dimensionless drag coefficient $c_d$ into account. The variation of the density of air can be neglected for the low vertical extent of the flight profiles, it is assumed to be constant ($\rho = 1.2 \ \mathrm{kg\,m^{-3}}$).

$$\mathbf{F_w} \quad = \quad \frac{\rho}{2} c_d A \mathbf{V}_w^2 \tag{12}$$

In a stable hover state, we assume that inertial forces on the left hand side of Eq. (9) can be neglected and thus

$$F_{w,x} \quad = \quad mg\sin\theta \tag{13}$$

By taking Eq. (12) into account, Eq. (13) leads to the following equation for wind speed in the direction the quadrotor is facing:

$$\frac{\rho}{2} c_d A V_{w,x}^2 \quad = \quad F_{w,x} \tag{14}$$

$$V_{w,x} \quad = \quad \sqrt{\frac{2F_{w,x}}{c_d A \rho}} \tag{15}$$





The term $c_d A$ is unknown and requires calibration. The drag coefficient and the projected area vary with the attitude of the quadrotor.

$$c_d A \quad = \quad c_{d,0} A_0 + f(\theta) \quad , \tag{16}$$

where $c_{d,0} A_0$ is the drag coefficient and area at zero pitch angle. In this study we assume that the function $f(\theta)$ is a linear function.

### 4.3 Hover state accuracy

In order to calculate the wind velocity with the introduced method, the UAVs have to maintain a hover state. In order to evaluate the validity of the assumption of negligible linear and angular motion, we can look at the variance of the measured positions from GNSS data. The mean horizontal standard deviation of horizontal movement over all 76 flights of the campaign for all quadrotors results in $\sigma_{p,h} = 0.19$ m. The vertical stability appears to be slightly lower with $\sigma_{p,v} = 0.85$ m. These measured standard deviations are within the accuracy of the GNSS measurement which is estimated to be of the order of $\epsilon_h = 0.6$ m in horizontal direction and $\epsilon_v = 0.8$ m in vertical direction by the avionic system. This means that actual movements can be slightly larger than the measured standard deviations, but are still very small. These findings are also confirmed by visual inspection of the flights.

## 5 Calibration of wind measurement

As mentioned in Sect. 4.2, the parameter $c_d A$ can not be estimated from system specifications alone. In order to calculate this parameter, calibration flights were performed at the GM Falkenberg during the FESST@MOL campaign as described in Sect. 3. In the following comparison the meteorological mast provides the reference for the wind measurements of the quadrotors. For the calibration only "drone tower" pattern flights were used. In total 34 flights with multiple quadrotors were performed in this pattern. As established previously in Eq. (16) the parameter $c_d A$ is approximated by a constant and a linear term depending on the pitch angle with the proportional parameter $c_p$ (Eq. 17).

$$c_d A \quad = \quad c_{d,0} A_0 + c_p * \theta \tag{17}$$

One flight consists of approx. 10 minutes of hovering, data were averaged over this time period for the following calibration steps in order to compare the data with corresponding 10-minute averaged wind speeds of the anemometers on the mast.

### 5.1 Individual quadrotor wind calibration

In the first step, each quadrotor is calibrated individually against the reference with all "drone tower" flights. Beside the determination of the parameters $c_{d,0} A_0$ and $c_p$, an offset for the pitch angle is introduced as $\Delta\theta$. This is necessary because of misalignment in the installation of the IMU in the quadrotor frame and slight differences in the mass distribution of the individual systems. Once the offset is determined it is applied to the measured pitch angle before any further processing.





The optimal calibration function is obtained by solving a defined non-linear least-squares problem. In particular bounds were defined and the minimization was performed by the Trust Region Reflective algorithm (Branch et al., 1999). The resulting wind speed for this calibration is plotted in Fig. 3. One single data point represents the time-averaged wind speed of a single flight of 235 one quadrotor in comparison to the corresponding average of the tower reference measurement. Due to some technical issues with quadrotor #4, it is not taken into consideration in the further calibration procedure. The root-mean-square deviations ($\sigma_{\mathrm{rms}}$) of the calculated wind speed against the reference as well as a bias ($\Delta V_w$) is determined from all single flights for the individual quadrotors and listed in the left column of Table 2. In the present case the calibration dataset is equal to the validation data, therefore the deviation is expected to be relatively small and remaining differences include the atmospheric 240 variability in mostly convective ABLs. For this calibration, the averaged bias between quadrotor wind speed measurements and the reference wind speed is $\Delta V_w < 0.01 \ \mathrm{m\,s^{-1}}$. The random deviation for time averaged data is $\sigma_{\mathrm{rms}} = 0.23 \ \mathrm{m\,s^{-1}}$ on average over all flights. The accuracies of each UAV are listed in Table 2. This kind of calibration with many flights in different conditions and individual coefficients for each UAV is considered the best possible calibration and the benchmark for more simplified calibration procedures with reduced calibration datasets and calibration parameters that are common for the whole swarm.

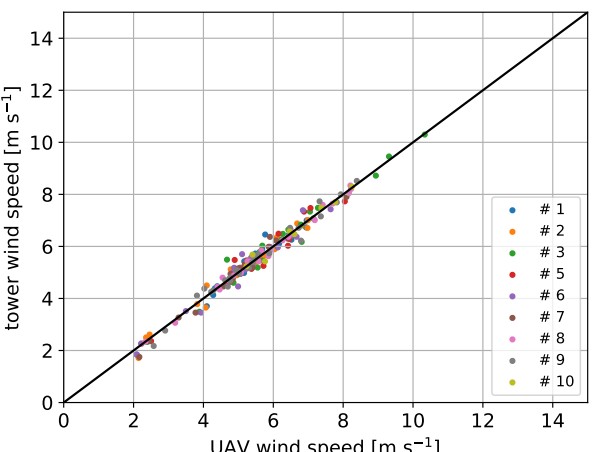

**Figure 3.** 10-minute averaged wind speed for n=34 flights drone vs. tower using the individual parameter calibration from the same 34 flights.


## 5.2 Aerodynamic calibration

The aim of the study is to implement a robust calibration for a large number of UAVs in a swarm. In a large number of drones it will not always be possible to obtain as many calibration flights. Quadrotors of the same built should however have very similar aerodynamic characteristics. In order to achieve this requirement, one common set of parameters $c_{\mathrm{d},0} A_0$ and $c_p$ with 250 sufficiently reasonable accuracy for all regarded quadrotors shall therefore be determined in this section. Only the pitch offset





**Table 2.** Accuracy of wind speed measurement in $[\mathrm{m\,s^{-1}}]$ for a dataset of 34 flights (used for calibration and validation) for a) calibration with all 3 parameters and b) using only pitch offset for calibration with universal parameter values for $c_p$ and $c_{\mathrm{d},0}A_0$.

| | | individual (Fig. 3) | | universal | |
|---|---|---|---|---|---|
| | # | $\Delta V_w$ | $\sigma_{\mathrm{rms}}$ | $\Delta V_w$ | $\sigma_{\mathrm{rms}}$ |
| | 1 | 0.01 | 0.24 | 0.01 | 0.26 |
| | 2 | 0.00 | 0.20 | 0.00 | 0.22 |
| | 3 | 0.00 | 0.27 | -0.01 | 0.29 |
| | 5 | 0.00 | 0.33 | 0.00 | 0.33 |
| | 6 | 0.01 | 0.28 | 0.03 | 0.29 |
| | 7 | 0.02 | 0.21 | 0.07 | 0.27 |
| | 8 | 0.00 | 0.12 | 0.00 | 0.13 |
| | 9 | 0.00 | 0.22 | 0.01 | 0.22 |
| | 10 | 0.00 | 0.17 | 0.01 | 0.21 |
| | mean | | 0.23 | | 0.25 |

remains as an individual calibration parameter for each quadrotor. By using the dataset of 34 flights the following function is obtained by taking the mean of the parameters of all quadrotors to minimize the overall error (Eq. (18)).

$$c_d A \quad = \quad 0.03 - 0.047 * \theta \tag{18}$$

Figure 4 demonstrates that the obtained curve fits well with the individual data points of all quadrotors. The result of the
calibration with common parameters is shown in the right column of Table 2. In comparison to the individual calibration, the accuracy is still reasonably high ($\sigma_{\mathrm{rms}} = 0.25$ m s$^{-1}$). The obtained value for $c_{\mathrm{d},0}A_0$ is in the range of the expected value as it would be calculated from an approximated surface area $A \approx 0.25 \cdot 0.05$ m$^2$ and the drag coefficient of a long flat plate $c_d = 2$. Estimation of the constant parameter $c_{\mathrm{d},0}A_0$ from these parameters leads to a value of 0.025 m$^2$. Of course, the drag coefficient and surface area of the quadrotor with running rotors cannot be measured this simply, which is why the calibration
is considered necessary. Comparing this result with the mentioned studies in literature (see Introduction) different functions were obtained for the relation between wind speed and quadrotor attitude. In our study the relation is more complex, but could roughly be described as a root function.

### 5.3 Pitch offset calibration

Having established aerodynamic parameters which appear to be universally applicable to the SWUF-3D swarm, it is still
necessary to calibrate the pitch offset $\Delta\theta$ for each individual quadrotor. In this section, we evaluate how many calibration flights are necessary and how stable the calibration is, i.e. how large the errors grow if only fewer calibration flights are used. First, the full dataset of 34 flights is used to determine $\Delta\theta$. Then, different calibration scenarios are performed with the present





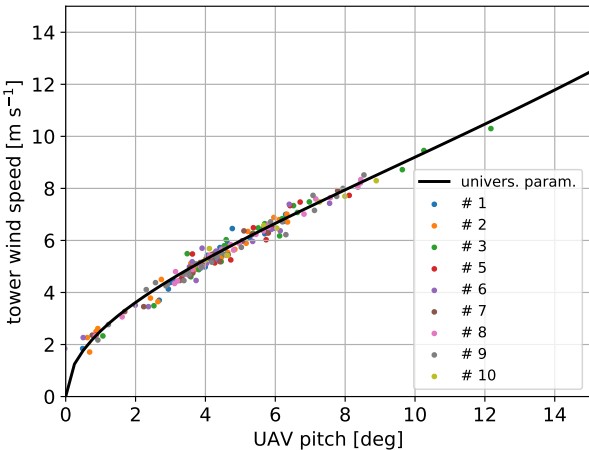

**Figure 4.** Regression function used for determining one universal set of parameters for $c_p$, $c_{d,0}A_0$.

dataset and the related RMS deviations in comparison to the tower measurements are calculated. In order to use a common validation dataset, only "drone tower" flights of the second week (see Table C2) are used to calculate RMS deviations and wind

speed bias. Four different calibration scenarios were performed:

a All 34 "drone tower" flights are used to estimate individual values of $\Delta\theta$. This should yield the best results for the accuracy estimation.

b Only the first week of flights is used for calibration, i.e. 22 "drone tower" flights as listed in Tab. C1. In this scenario, the calibration dataset is completely independent of the validation dataset, but still quite large, with a variety of wind

conditions.

c In order to simplify the calibration and evaluate if $\Delta\theta$ is stable throughout the whole campaign, only one flight is considered for calibration. Flight #31 is selected as the calibration flight as it has an average wind speed of $6\,\mathrm{m\,s^{-1}}$. The pattern drone tower is only performed with eight quadrotors, which is why UAV#10 is not included in this calibration.

d For the calibration of all quadrotors in the swarm, a second calibration flight is required. Flight #31 and #56 are used as

data base in the following sections for calculating the wind speed (Fig. 5).

The accuracy estimates of the respective calibration scenarios are listed in Table 3. It is found that reducing the number of calibration flights yields lower accuracy, as expected. Using only the first week as the calibration dataset increases the averaged $\sigma_{\mathrm{rms}}$ from $0.23\,\mathrm{m\,s^{-1}}$ to $0.25\,\mathrm{m\,s^{-1}}$ and using only a single flight increases it further to $0.28\,\mathrm{m\,s^{-1}}$. Both, bias and RMS deviation increase if less flights are used to estimate $\Delta\theta$, which suggests that the offset is not completely stable throughout the

campaign. However, even the largest deviation estimate of quadrotor #5 is still below $0.5\,\mathrm{m\,s^{-1}}$, which is considered acceptable for this study, but it will be a goal to improve this in future. The calibrated pitch offset parameter ranges in between $\pm4.3°$.



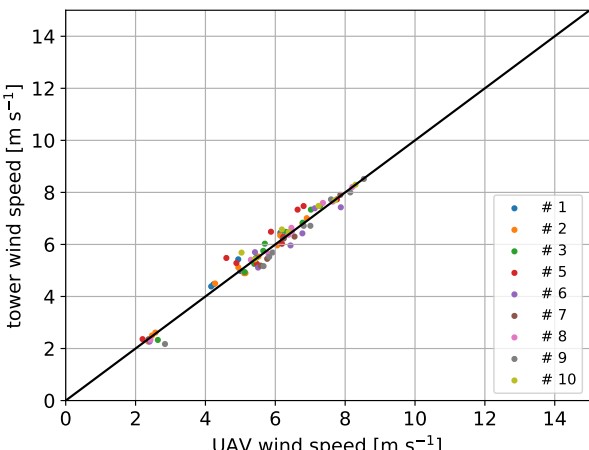

**Figure 5.** Average wind-speed for n=12 flights from the second week drone vs. tower using the universal parameter - only individual pitch offset is calibrated from flight number 31 and 56 (scenario (d)).

**Table 3.** Accuracy of wind speed measurements in $[\mathrm{m\,s^{-1}}]$ for different calibration data using only pitch offset applied on validation dataset of 12 flights from the second week.

| # | (a) n34 all | | (b) n12 first week | | (c) n1 fl.31 | | (d) n2 fl.31+56 (Fig. 5) | |
|---|---|---|---|---|---|---|---|---|
|  | $\Delta V_w$ | $\sigma_{\mathrm{rms}}$ | $\Delta V_w$ | $\sigma_{\mathrm{rms}}$ | $\Delta V_w$ | $\sigma_{\mathrm{rms}}$ | $\Delta V_w$ | $\sigma_{\mathrm{rms}}$ |
| 1 | 0.00 | 0.16 | 0.00 | 0.16 | -0.21 | 0.27 | -0.21 | 0.27 |
| 2 | -0.13 | 0.20 | -0.25 | 0.30 | -0.24 | 0.29 | -0.01 | 0.15 |
| 3 | -0.12 | 0.23 | -0.19 | 0.27 | -0.11 | 0.22 | -0.06 | 0.21 |
| 5 | -0.04 | 0.37 | -0.19 | 0.41 | -0.28 | 0.47 | -0.31 | 0.48 |
| 6 | -0.08 | 0.32 | -0.12 | 0.33 | 0.18 | 0.35 | 0.18 | 0.35 |
| 7 | 0.09 | 0.19 | 0.11 | 0.20 | 0.13 | 0.21 | 0.16 | 0.23 |
| 8 | -0.02 | 0.12 | -0.03 | 0.12 | -0.03 | 0.12 | -0.03 | 0.12 |
| 9 | -0.05 | 0.22 | -0.06 | 0.22 | 0.25 | 0.34 | 0.23 | 0.33 |
| 10 | 0.03 | 0.23 | 0.13 | 0.26 | - | - | -0.20 | 0.30 |
| mean |  | 0.23 |  | 0.25 |  | 0.28 |  | 0.27 |

## 5.4 Yaw-offset determination

Additionally to the magnitude of the wind speed, the wind direction is obtained from the quadrotors. As mentioned in Sect. 2, the quadrotors were operated in weather-vane mode. Hence, the quadrotor is automatically yawed in the direction of the wind. 290 Therefore, the corresponding wind direction can directly be obtained from the yaw angle $\psi$. Nevertheless, the magnetometer





is not always perfectly orientated towards north and calibration deviations can occur, which makes an offset calibration of the yaw angle necessary. For this purpose, the same two flights as for the wind speed calibration are used (i.e. flight #31 and #56). The calibrated average wind direction for the "drone tower" flights of the second week is plotted in Fig. 6. The mean RMS deviation results in $\sigma_{\mathrm{rms},\psi} = 7.5°$. The few outliers can be explained by low wind speed conditions, when roll angles above $1°$ are hardly reached and the weather-vane mode does not always correct the yaw angle fully into the wind direction.

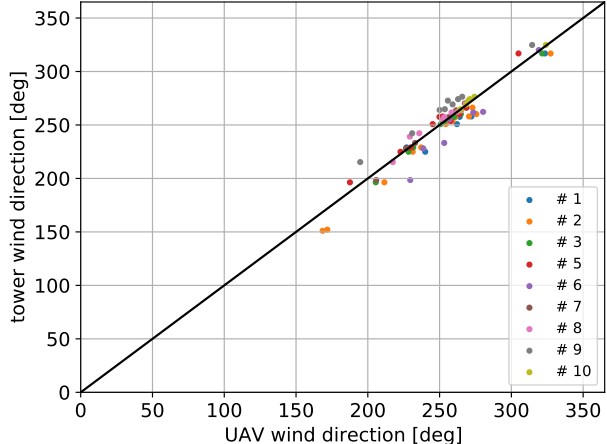

**Figure 6.** Average wind direction for $n$=12 flights from the second week drone vs. tower including offset calibration.


## 6  Validation of synchronous swarm measurements

The goal of the SWUF-3D swarm is to capture small- to meso-scale flow structures in the ABL. Having calibrated the quadrotors for good wind measurement accuracy, we now evaluate how the synchronous measurements of multiple drones compare to synchronous measurements of multiple anemometers on the 99 m mast and with Doppler lidar wind measurements.

### 6.1  Vertical profiles

The "drone tower" flight pattern which was also used for calibration is suitable for measurements of vertical profiles with the quadrotors. As an example, we present flights #61 and #62 (without UAV #4) of the campaign since they feature shear and some gustiness in the wind field and were performed in close succession. In Fig. 7a, only the quadrotor at 90 m is compared to the corresponding sonic measurement at the same height. It is evident from this plot that not only the 10-minute averaged

values are in good agreement with the reference instruments, but also the resolved time series of wind speed matches the sonic anemometer data very well. The variance of the velocity fluctuation of the 1 Hz data of the quadrotor $\sigma_{\mathrm{v,q}}^2 = 1.76 \ \mathrm{m^2 s^{-2}}$ is thus in good agreement with the sonic data $\sigma_{\mathrm{v,s}}^2 = 1.65 \ \mathrm{m^2 s^{-2}}$ for this particular case. Figures 7b and 7c show the time series of the vertical profile for cup anemometers on the mast and seven quadrotors respectively. The data from the cup anemometers



are only available as 1-minute average values, which is why the complete met-mast data is shown in the contour plot with
a resolution of one minute and thus significantly less structure in the flow field is seen compared to the SWUF-3D swarm.
Nevertheless, periods of higher wind speeds and stronger shear are present and match well between SWUF-3D and mast. To
show the shear profile more clearly, Fig. 8 gives the averaged vertical profiles for the two ten-minute periods. The differences
between UAVs and mast measurements are of the same order as the previously determined RMS deviations.

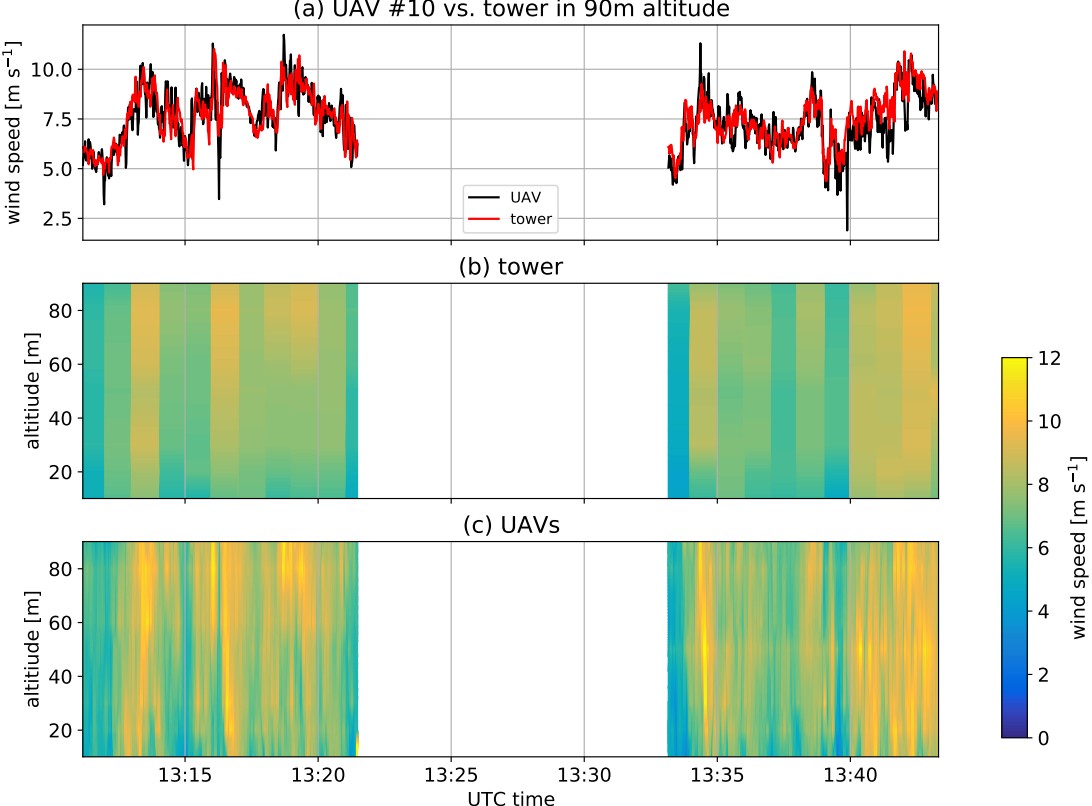

**Figure 7.** Comparison of drone tower to reference data from the tower for flight #61 and #62: a) one single UAV at 90 m altitude vs. sonic
measurements b) time series of tower measurements at different altitudes c) corresponding UAV time series of seven quadrotors at different
altitudes.

## 6.2   Variance

With the 1 Hz resolution of the quadrotor measurements, a significant part of the turbulent fluctuations can theoretically be
resolved. In order to evaluate the capability to measure wind speed variance, we compare all flights at 50 m and 90 m with the





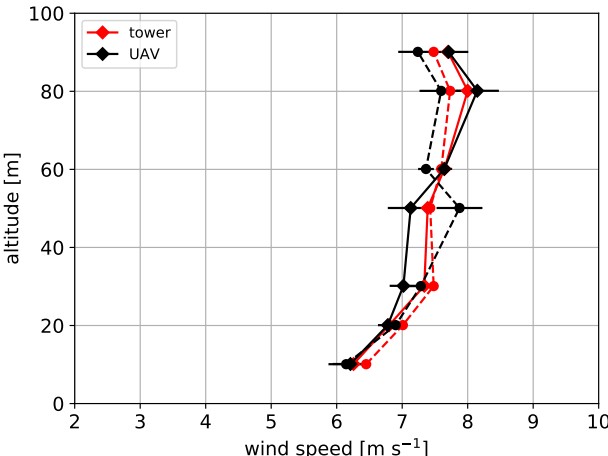

**Figure 8.** Comparison of drone tower to reference data from the tower for flight #61 (solid line) and #62 (dashed line). The error bars represent the RMS deviation that was determined for each individual quadrotor in Sect. 5.3.

corresponding sonic anemometers. Figure 9 shows the scatterplot of this comparison. The mean RMS deviation of the variance is $\sigma_{\mathrm{rms},\sigma^2} = 0.37 \ \mathrm{m^2 s^{-2}}$. Given the convective nature of the ABL in which most of the flights were performed, we consider the agreement satisfactory. Further detailed analysis of the scales that are resolved with the quadrotor are out of the scope of this study and will be handled in a separate study.

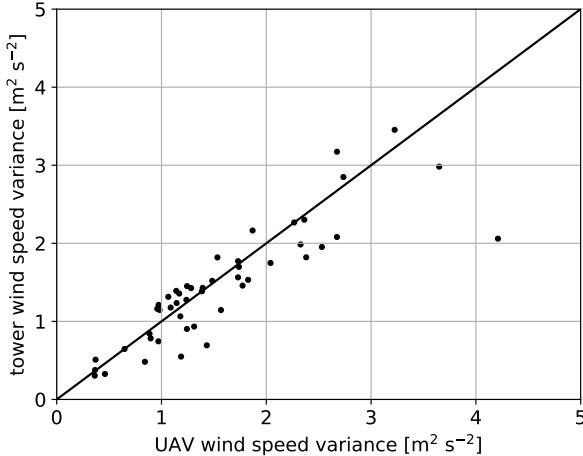

**Figure 9.** Variance of the wind-speed for n=34 flights in comparison to sonic measurements at 50 m and 90 m altitude.




## 6.3 Horizontal line

The long-range lidar was used for further validation of the possibility to resolve horizontal structures in the atmospheric boundary layer with the SWUF-3D swarm. In one scenario, the lidar was set to measure continuously at a fixed elevation (7°) and azimuth angle (280°). Eight quadrotors were placed along the line-of sight in the same spacing as the range gate separation

of the lidar, which was set to 20 m with the closest range gate 80 m from the lidar (see also Fig. 2). As the lidar is measuring with a non-zero elevation angle, there is a height difference of 18 m between the position of the first range gate at 10 m above ground level (agl) and the last range gate at 28 m agl. Figure 10 shows the comparison of the time series of the interpolated horizontal line. It is evident how both measurement systems measure the same variations in wind speed. A significant gust occurred at 13:27 UTC for example, which is observed with both systems. The lidar measurements show smoother gradients

in wind speed variations along the line of sight which can be attributed to the volume averaging effect that is inherent to the method and can also explain the lower maximum values of the lidar measurements.

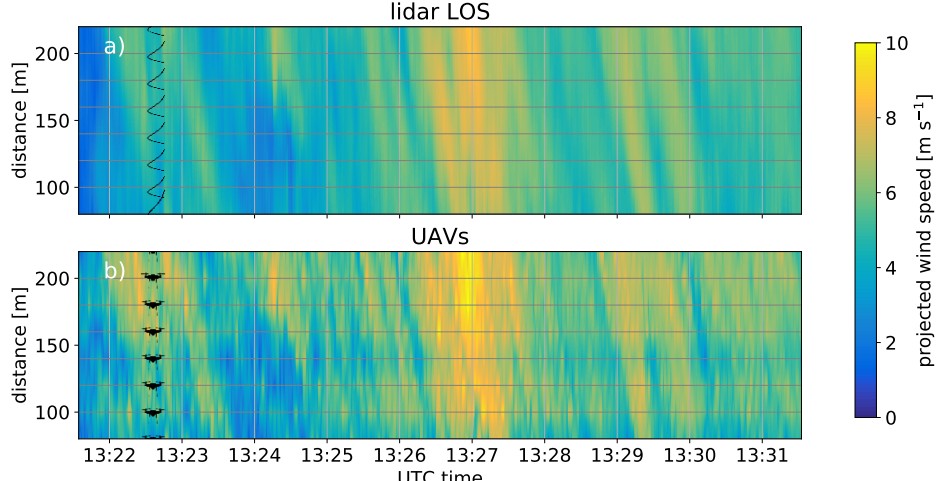

**Figure 10.** Time series of spatial wind measurements with a lidar (a) and eight drone (b) along a lidar line-of-sight. The y-axis grid represents the location of the range gate centers of the lidar and the drones respectively.

## 6.4 Vertical plane

In order to evaluate the performance of the UAV swarm to measure wind fields and their spatial distribution, flights were performed in a 3x3 grid, in the measurement plane of a lidar RHI (range-height indicator) scan. Figure 11 shows the resulting

time series of one ten-minute flight. The lidar data is linearly interpolated to the quadrotor location and the horizontal wind component is reconstructed by division through the cosine of the elevation angle ($v_h = \frac{v_r}{\cos \phi}$). The 1 Hz quadrotor data points are centered at the time when the lidar beam crossed the quadrotor location and the wind component in line with the RHI



plane was calculated from quadrotor wind speed and wind direction. It shows that the main features of the flow are captured
similarly with quadrotor and lidar. At the location of the central quadrotor, the lidar showed some hard target reflections that
were probably caused by the quadrotor and lead to some gaps in the data for this location. As for the previous validation
measurements, a good agreement to the reference system is found with deviations between quadrotor and lidar that are of the
order of the previously determined accuracies. This example gives some confidence that spatial structures can be well captured
with the SWUF-3D swarm even though the convective nature of the ABL in this experiment is extremely challenging for a
direct comparison to the reference instruments.

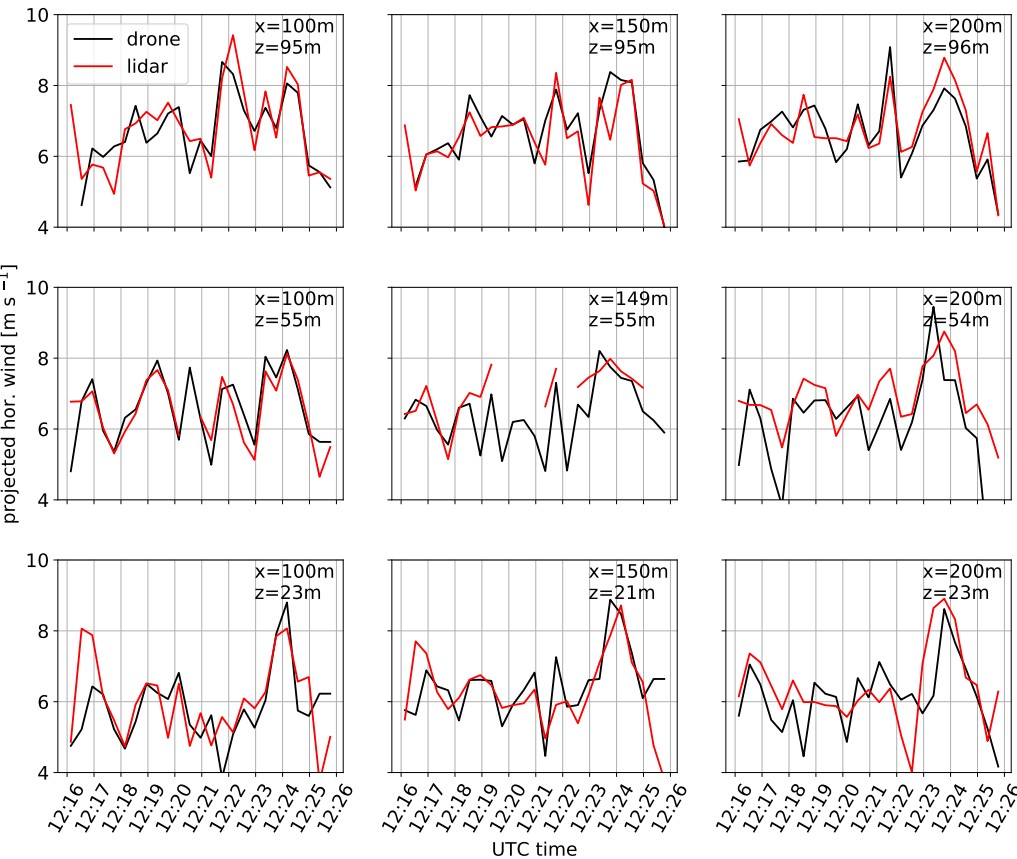

**Figure 11.** Time series of wind measurements of nine quadrotors compared to corresponding lidar measurements at the same locations on
24 July 2020.





# 7 Conclusions

Atmospheric measurements with multirotor UAVs are of increasing interest to the scientific community because of the many new possibilities for flexible measurements with a quick and low-cost deployment. In order to establish the technology and classify the quality of the measurements, a transparent description of the algorithms and a traceable validation is important. In this study we described an algorithm for wind measurements that is based on the physical principle of aerodynamic drag and the related quadrotor dynamics. With the goal to enable swarm measurements that can capture small-scale structures in the ABL, nine quadrotors were calibrated against wind measurements of sonic and cup anemometers installed on the 99-m mast at the GM Falkenberg. An overall accuracy of $\sigma_{rms} < 0.3 \, \mathrm{m \, s^{-1}}$ for the wind speed and $\sigma_{\mathrm{rms},\psi} < 8°$ for wind direction measurement was found. The SWUF-3D swarm is then successfully validated using lidar and mast measurements. The major achievements of the study can be summarized as follows:

- A commercial racing drone was utilized as a measurement system. The choice of this kind of UAV proofed to be very beneficial, since the dynamics of the small quadrotors allow for a sensitive calibration curve. Also, the stability of the hover position is important for the measurement of turbulent winds and the systems can operate in high wind speeds.

- The algorithm is successfully calibrated for individual quadrotors resulting in an average accuracy of $\sigma_{rms} = 0.23 \, \mathrm{m \, s^{-1}}$ if a large number of calibration flights is used.

- One universal set of aerodynamic parameters is determined for the whole swarm. An accuracy of wind measurements as high as $\sigma_{rms} = 0.27 \, \mathrm{m \, s^{-1}}$ is achieved although only two flights were taking into account for the calibration of pitch misalignment offsets. This leads to the possibility of fast swarm calibration by using only few flights, which should however be chosen to be performed in medium to high wind speeds. RMS deviation includes the uncertainty due to the location offset between quadrotors and sonics that was comparably large ($\approx 20 \, m$) in this study. The atmospheric variability can be especially large since all flights were performed during daytime, mostly in a well-developed convective ABL.

- The application of a weather-vane mode simplifies both, the measurement of wind speeds and wind direction. The wind speed measurement algorithm can thus be reduced to a pitch-angle relationship and wind direction measurements can be directly read from the yaw angle of the UAV.

- Lidar and tower comparison show that detailed flow structures both in time and in space could be resolved with the quadrotors. In the given configuration, the quadrotor data have a higher spatial resolution than the long-range lidar data and allow to detect turbulent structures like e.g. wind gusts.

# 8 Outlook

In future, further analysis of the data and improvement of the wind algorithm will be considered. Some of the major fields of future research and development are:





– Improving the algorithm of wind measurement by increasing the level of complexity, i.e. for example to dissolve the assumptions that were made for the hover state by taking gyroscopic terms into account. Also, the roll angle could be included to resolve small-scale disturbances which are not in line with the main wind direction. Making use of the available information of motor output could potentially allow even finer resolution and vertical wind measurements, but needs significantly improved system identification and calibration.

– Analyzing measurement data towards turbulence intensity, correlation between multiple UAVs, coherence and turbulent structures in general. A big advantage and goal of the SWUF-3D swarm is to analyze turbulence without the assumption of Taylor's hypothesis of frozen turbulence as it is usually necessary with airborne measurements or stationary mast measurements. The swarm with multiple measurements in space can potentially directly measure cross-correlation and structure functions in space.

– The simultaneous data of the SWUF-3D swarm can be a very valuable tool for validation of numerical simulations such as large-eddy simulations. In future, comparisons to such models will be pursued.

– Improvement of flight time with higher battery capacity and controller optimization. Flight times of 17 minutes were reached in some test flights with the presented quadrotors under best conditions. Significantly longer flight times would however require larger UAVs.

– Expanding the SWUF-3D swarm up to 100 quadrotors for a larger grid of wind measurements. It is the goal for SWUF-3D to measure turbulent structures in the wake of wind turbines. With the results of this study, it will be the next step to fly in close vicinity to wind turbines.

– Improving the temperature and humidity measurements of the quadrotors. It was found, although not presented in this study, that the sensors were installed to close too the body of the quadrotor and suffered from radiative heating of the system itself. An improved installation will solve this problem in future.

*Data availability.* The data are available from the author upon request.





## Appendix A:  Nomenclature

| | |
|---|---|
| $\epsilon_h$ | horizontal accuracy of the GNSS measurement [m] |
| $\epsilon_v$ | vertical accuracy of the GNSS measurement [m] |
| $\rho$ | density of air [$\mathrm{kg\,m^{-3}}$] |
| $\sigma_{\mathrm{p,h}}$ | standard deviation of the measured horizontal position [m] |
| $\sigma_{\mathrm{p,v}}$ | standard deviation of the measured vertical position [m] |
| $\sigma_{\mathrm{v,q}}^2$ | variance of the velocity fluctuation for the quadrotor measurement [$\mathrm{m^2 s^{-2}}$] |
| $\sigma_{\mathrm{v,s}}^2$ | variance of the velocity fluctuation for the sonic measurement [$\mathrm{m^2 s^{-2}}$] |
| $\sigma_{\mathrm{rms}}$ | root-mean-square deviations of wind speed calculation [$\mathrm{m\,s^{-1}}$] |
| $\sigma_{\mathrm{rms},\sigma^2}$ | root-mean-square deviation of the variance measurement [$\mathrm{m^2 s^{-2}}$] |
| $\sigma_{\mathrm{rms},\psi}$ | root-mean-square deviations of wind direction calculation [-] |
| $\omega_{\mathbf{b}}$ | angular velocity vector in body frame [$\mathrm{s^{-1}}$] |
| $\Delta\theta$ | pitch angle offset of quadrotor measurement [°] |
| $\Delta V_w$ | bias of wind speed calculation [$\mathrm{m\,s^{-1}}$] |
| $\mathbf{\Phi_i}$ | angular vector in inertial frame [$\mathrm{rad}$] |
| $\phi$ | roll angle of quadrotor [$\mathrm{rad}$] |
| $\theta$ | pitch angle of quadrotor [$\mathrm{rad}$] |
| $\psi$ | yaw angle of quadrotor [$\mathrm{rad}$] |
| $c_d$ | drag coefficient [-] |
| $c_{d,0}$ | drag coefficient for zero pitch angle [-] |
| $c_p$ | proportional parameter for aerodynamic drag calibration [$\mathrm{m^2}$] |
| $d$ | thrust coefficient [$\mathrm{m\,kg}$] |
| $g$ | acceleration due to gravity [$\mathrm{m\,s^{-2}}$] |
| $m$ | mass [$\mathrm{kg}$] |
| $n_i$ | rotational speeds of motors [$\mathrm{s^{-1}}$] |
| $A$ | projected front area [$\mathrm{m^2}$] |
| $A_0$ | projected front area for zero pitch angle [$\mathrm{m^2}$] |
| $\mathbf{F_c}$ | external forces [N] |
| $\mathbf{F_e}$ | control forces [N] |
| $\mathbf{F_g}$ | gravity force [N] |
| $F_{w,x}$ | wind forces in x-direction [N] |
| $\mathbf{I}$ | rotational inertia [$\mathrm{kg\,m^2}$] |
| $\mathbf{M_c}$ | control moments [N m] |
| $\mathbf{M_e}$ | external moments [N m] |
| $\mathbf{R}$ | rotation matrix [-] |
| $\mathbf{V_b}$ | translation velocity vector in body frame [$\mathrm{m\,s^{-1}}$] |
| $V_w$ | wind velocity [$\mathrm{m\,s^{-1}}$] |
| $\mathbf{X_i}$ | position vector in inertial frame [m] |





## Appendix B: Transformation Matrix

Rotation matrix $\mathbf{R}(\mathbf{\Phi_i})$:

$$\mathbf{R}(\mathbf{\Phi_i}) = \begin{bmatrix} \cos\theta\cos\psi & \cos\psi\sin\theta\sin\phi - \cos\phi\sin\psi & \cos\psi\sin\theta\cos\phi + \sin\phi\sin\psi \\ \cos\theta\sin\psi & \cos\phi\cos\psi + \sin\theta\sin\phi\sin\psi & -\sin\phi\cos\psi + \sin\theta\cos\phi\sin\psi \\ -\sin\theta & \cos\theta\sin\phi & \cos\theta\cos\phi \end{bmatrix} \tag{B1}$$

## Appendix C: Flight protocol

*Author contributions.* TW developed the algorithm to measure wind with the quadrotor, conducted the experiment and performed the data
analysis. NW analyzed the comparison to the lidar measurements. FB provided the tower measurements and wrote the section about the
FESST@MOL experiment. TW wrote the paper, with significant contributions from NW. All co-authors contributed to refining the paper
text.

*Competing interests.* The authors declare that they have no competing interests.

*Acknowledgements.* We thank Annika Gaiser and Felix Schmitt for their assistance in the FESST@MOL experiment. Martin Hagen inter-
nally reviewed the manuscript and we thank him for his valuable comments.
Pixhawk is a trademark of Lorenz Meier.
The source code of QGroundControl is dual-licensed under Apache 2.0 and GPLv3 (or later), the artwork/images are licensed under CC by
SA. Except where otherwise noted, content on this site is licensed under the following license: CC Attribution-Share Alike 3.0 Unported.




**Table C1.** Flight protocol of FESST@MOL campaign first week.

| Date | no | time start utc | time end utc | 1 | 2 | 3 | 4 | 5 | 6 | 7 | 8 | 9 | 10 | flight pattern | wind speed 98m [ms⁻¹] | wind dir. 98m [deg] | temp 98m [°C] | hum 98m [%] |
|---|---|---|---|---|---|---|---|---|---|---|---|---|---|---|---|---|---|---|
| 20.7 | 1 | 09:14:50 | 09:23:50 | x | x |  |  |  | x | x |  |  |  | drone tower | 1.14 | 216 | 22.2 | 77.3 |
| 20.7 | 2 | 10:02:30 | 10:11:40 | x | x |  |  |  | x | x |  |  |  | drone tower | 5.57 | 314 | 22.95 | 69.8 |
| 20.7 | 3 | 10:18:30 | 10:27:50 | x | x |  |  |  | x | x |  |  |  | drone tower | 5.23 | 320 | 23.15 | 67.3 |
| 20.7 | 4 | 11:30:10 | 11:39:20 | x | x |  |  |  | x | x |  |  |  | drone tower | 5.58 | 335 | 23.46 | 63.9 |
| 20.7 | 5 | 11:45:10 | 11:54:10 | x | x |  |  |  | x | x |  |  |  | drone tower | 5.94 | 337 | 23.54 | 61.9 |
| 20.7 | 6 | 13:47:00 | 13:56:00 |  |  | x | x |  |  |  | x | x |  | drone tower | 9.98 | 305 | 22.54 | 55.2 |
| 20.7 | 7 | 14:01:40 | 14:10:50 |  |  | x | x |  |  |  | x | x |  | drone tower | 9.32 | 313 | 22.42 | 58.3 |
| 20.7 | 8 | 14:19:40 | 14:28:50 |  |  | x | x |  |  |  | x | x |  | drone tower | 9.89 | 315 | 22.51 | 57.9 |
| 21.7 | 9 | 08:49:10 | 08:58:00 |  |  | x | x |  |  |  | x | x |  | drone tower | 4.49 | 289 | 17.23 | 58.9 |
| 21.7 | 10 | 09:56:20 | 10:05:10 |  |  | x | x |  |  |  | x | x |  | drone tower | 4.88 | 280 | 17.97 | 50.5 |
| 21.7 | 11 | 10:11:20 | 10:20:00 |  |  | x | x |  |  |  | x | x |  | drone tower | 5.87 | 300 | 17.78 | 47.5 |
| 21.7 | 12 | 12:35:30 | 12:41:45 |  |  | x | x |  |  |  | x | x |  | drone tower | 6.79 | 305 | 19.04 | 37.8 |
| 21.7 | 13 | 13:07:00 | 13:13:20 |  |  | x | x |  |  |  | x | x |  | drone tower | 6.81 | 290 | 19.5 | 32.2 |
| 21.7 | 14 | 13:20:30 | 13:29:20 |  |  | x | x |  |  |  | x | x |  | drone tower | 5.91 | 299 | 19.77 | 32.1 |
| 21.7 | 15 | 14:31:40 | 14:41:00 | x | x | x | x |  | x | x | x | x |  | drone tower | 6.28 | 307 | 19.97 | 31.5 |
| 21.7 | 16 | 14:56:40 | 15:04:00 | x | x | x | x |  | x | x | x | x |  | drone tower | 6.81 | 298 | 20.19 | 32 |
| 22.7 | 17 | 09:06:30 | 09:15:30 | x | x | x | x |  | x | x | x | x |  | drone tower | 5.66 | 289 | 16.7 | 49.6 |
| 22.7 | 18 | 09:22:30 | 09:31:00 | x | x | x | x |  | x | x | x | x |  | drone tower | 6.72 | 288 | 16.85 | 47 |
| 22.7 | 19 | 11:04:40 | 11:13:30 |  |  | x | x |  |  |  | x | x |  | 1x4 lidar | 5.28 | 303 | 17.81 | 39.4 |
| 22.7 | 20 | 11:18:10 | 11:27:30 |  |  | x | x |  |  |  | x | x |  | 1x4 lidar | 5.05 | 298 | 18.06 | 39.2 |
| 22.7 | 21 | 12:14:40 | 12:22:00 | x | x | x | x |  | x | x | x | x |  | 2x4 lidar | 5.59 | 300 | 18.7 | 36.9 |
| 22.7 | 22 | 12:40:50 | 12:49:10 | x | x | x | x |  | x | x | x | x |  | 2x4 lidar | 5.01 | 294 | 18.8 | 35.9 |
| 22.7 | 23 | 14:57:00 | 15:05:30 | x | x | x | x | x | x | x | x | x | x | DLR logo | 5.12 | 307 | 19.58 | 35.3 |
| 23.7 | 24 | 10:35:20 | 10:43:50 | x | x | x | x |  | x | x | x | x |  | drone tower | 2.72 | 311 | 17.5 | 53.5 |
| 23.7 | 25 | 11:21:00 | 11:28:30 | x | x | x | x |  | x | x |  | x | x | drone line | 1.81 | 257 | 17.56 | 52.5 |
| 23.7 | 26 | 13:03:40 | 13:12:00 | x | x | x | x |  | x | x |  | x | x | 2x4 lidar | 3.42 | 253 | 18.81 | 44.7 |
| 23.7 | 27 | 13:24:40 | 13:33:20 | x | x | x | x |  | x | x |  | x | x | 2x4 lidar | 4.04 | 254 | 19.12 | 41.3 |
| 24.7 | 28 | 07:45:40 | 07:55:00 | x | x | x | x |  | x | x |  | x | x | 2x4 lidar | 4.36 | 210 | 20.12 | 47.6 |
| 24.7 | 29 | 08:04:50 | 08:08:20 | x | x | x | x |  | x | x |  | x | x | 2x4 lidar | 4.84 | 221 | 20.42 | 45.5 |
| 24.7 | 30 | 09:59:20 | 10:09:30 | x | x | x |  | x | x | x | x |  |  | drone tower | 5.42 | 251 | 22.76 | 35.5 |
| **24.7** | **31** | **10:19:30** | **10:29:10** | x | x | x |  | x | x | x | x | x |  | **drone tower** | **6.51** | **209** | **23.05** | **34.4** |
| 24.7 | 32 | 11:52:20 | 12:01:50 | x | x | x |  | x | x | x | x | x | x | 3x3 lidar | 5.83 | 235 | 24.03 | 33.7 |
| **24.7** | **33** | **12:15:50** | **12:25:30** | x | x | x |  | x | x | x | x | x | x | **3x3 lidar** | **6.07** | **230** | **24.37** | **34.1** |
| 24.7 | 34 | 13:18:20 | 13:27:50 | x | x | x |  | x | x | x | x | x | x | 3x3 lidar | 5.96 | 261 | 24.87 | 31.9 |
| 24.7 | 35 | 14:14:00 | 14:24:00 | x | x | x |  | x | x | x | x | x | x | 3x3 lidar | 7.91 | 252 | 25.83 | 28.7 |
| 24.7 | 36 | 15:06:10 | 15:15:50 | x | x | x |  | x | x | x | x | x | x | 3x3 lidar | 7.21 | 272 | 25.9 | 28.4 |
| 24.7 | 37 | 15:53:50 | 16:03:50 | x | x | x |  | x | x | x | x |  | x | drone tower | 5.07 | 279 | 26.02 | 29.8 |





**Table C2.** Flight protocol of FESST@MOL campaign second week.

| Date | no | time start utc | time end utc | 1 | 2 | 3 | 4 | 5 | 6 | 7 | 8 | 9 | 10 | flight pattern | wind speed 98m [ms⁻¹] | wind dir. 98m [deg] | temp 98m [°C] | hum 98m [%] |
|---|---|---|---|---|---|---|---|---|---|---|---|---|---|---|---|---|---|---|
| | | | | | | | | qav no | | | | | | | | | | |
| 27.7 | 38 | 07:33:40 | 07:44:30 | | x | | x | | | | | | | drone tower | 2.23 | 137 | 18.66 | 78.3 |
| 27.7 | 39 | 07:49:20 | 08:00:00 | | x | | x | | | | | | | drone tower | 2.5 | 134 | 19.32 | 75.3 |
| 27.7 | 40 | 09:20:30 | 09:30:30 | | x | x | x | x | x | x | x | x | | drone tower | 3.04 | 186 | 21.17 | 63 |
| 27.7 | 41 | 13:24:30 | 13:33:30 | | | | | x | x | | | | | drone line | 2.26 | 182 | 23.82 | 41.2 |
| 27.7 | 42 | 13:45:10 | 13:55:00 | x | x | x | x | x | x | x | x | x | x | drone line | 2.76 | 198 | 23.88 | 42.7 |
| 28.7 | 43 | 07:21:40 | 07:31:20 | x | x | x | | x | x | x | x | x | | drone tower | 5.1 | 216 | 23.12 | 52.8 |
| 28.7 | 44 | 07:57:30 | 08:07:20 | x | x | x | | x | x | x | x | x | | drone tower | 5.86 | 235 | 23.79 | 52.3 |
| 28.7 | 45 | 08:41:20 | 08:48:50 | x | x | x | | x | x | x | x | x | | 2x4 lidar | 6.25 | 285 | 24.07 | 49.1 |
| 28.7 | 46 | 09:14:40 | 09:24:40 | x | x | x | | x | x | x | x | x | | 2x4 lidar | 4.88 | 291 | 23.73 | 50.4 |
| 28.7 | 47 | 11:07:10 | 11:16:30 | x | x | x | x | x | x | x | x | x | x | drone line | 4.77 | 222 | 25.46 | 44.4 |
| 28.7 | 48 | 11:24:30 | 11:33:30 | x | x | x | x | x | x | x | x | x | x | drone line | 6.07 | 248 | 25.56 | 43.4 |
| 28.7 | 49 | 13:03:10 | 13:12:20 | x | x | x | x | x | x | x | x | x | x | drone line | 8.73 | 291 | 25.14 | 44 |
| 28.7 | 50 | 13:24:40 | 13:34:30 | x | x | x | x | x | x | x | x | x | x | drone line | 5.94 | 291 | 25.1 | 44.8 |
| 28.7 | 51 | 14:45:50 | 14:54:30 | x | x | x | x | x | x | x | x | x | x | drone line | 6.71 | 266 | 25.94 | 39.4 |
| 28.7 | 52 | 15:14:30 | 15:23:30 | x | x | x | x | x | x | x | x | x | x | drone line | 6.43 | 295 | 25.38 | 41.8 |
| 29.7 | 53 | 07:16:50 | 07:26:20 | x | x | x | x | x | x | x | x | x | x | drone line | 5.96 | 291 | 16.7 | 56.2 |
| 29.7 | 54 | 07:41:50 | 07:51:10 | x | x | x | x | x | x | x | x | x | x | drone line | 5.74 | 291 | 16.63 | 56.9 |
| 29.7 | 55 | 08:45:40 | 08:55:10 | | x | x | x | x | | x | x | x | x | drone tower | 6.02 | 268 | 17.34 | 54.9 |
| **29.7** | **56** | **09:16:30** | **09:26:30** | | **x** | **x** | **x** | **x** | | **x** | **x** | **x** | | **drone tower** | **7.18** | **259** | **17.96** | **50.9** |
| 29.7 | 57 | 09:46:00 | 09:56:20 | | x | x | x | x | | x | x | x | | drone tower | 6.55 | 260 | 18.32 | 45.2 |
| 29.7 | 58 | 11:16:30 | 11:26:10 | x | x | x | | x | x | x | x | x | x | 3x3 lidar | 5.65 | 271 | 19.43 | 39.9 |
| 29.7 | 59 | 11:35:20 | 11:45:20 | x | x | x | | x | x | x | x | x | x | 3x3 lidar | 6.3 | 268 | 19.87 | 40.2 |
| 29.7 | 60 | 12:34:30 | 12:44:40 | x | x | x | x | | x | | x | x | x | drone tower | 8.32 | 287 | 20.07 | 40 |
| **29.7** | **61** | **13:11:10** | **13:21:30** | **x** | **x** | **x** | **x** | | **x** | | **x** | **x** | **x** | **drone tower** | **7.41** | **268** | **20.78** | **36.7** |
| **29.7** | **62** | **13:33:10** | **13:43:20** | **x** | **x** | **x** | **x** | | **x** | | **x** | **x** | **x** | **drone tower** | **9.03** | **268** | **20.99** | **36.1** |
| 29.7 | 63 | 14:34:00 | 14:44:00 | x | x | x | | x | x | x | x | x | x | 3x3 lidar | 7.12 | 261 | 21.42 | 35.2 |
| 29.7 | 64 | 15:09:20 | 15:19:10 | x | x | x | x | x | x | x | x | x | x | drone line | 7.12 | 276 | 21.38 | 34.3 |
| 30.7 | 65 | 07:33:20 | 07:43:30 | x | x | x | | x | x | x | x | x | x | 3x3 lidar | 4.23 | 282 | 15.99 | 56.9 |
| 30.7 | 66 | 07:55:50 | 08:05:40 | x | x | x | | x | x | x | x | x | x | 3x3 lidar | 4.46 | 279 | 16.3 | 55.3 |
| 30.7 | 67 | 09:43:20 | 09:53:00 | x | x | x | x | x | x | x | x | x | x | drone line | 5.6 | 266 | 18.04 | 48.5 |
| 30.7 | 68 | 10:03:10 | 10:12:50 | x | x | x | x | x | x | x | x | x | x | drone line | 5.51 | 252 | 18.12 | 48 |
| 30.7 | 69 | 11:45:10 | 11:55:10 | x | x | x | x | x | x | x | x | x | x | drone line | 4.74 | 279 | 20.02 | 40.4 |
| 30.7 | 70 | 12:12:20 | 12:22:20 | x | x | x | x | x | x | x | x | x | x | drone line | 4.75 | 307 | 20.25 | 39.4 |
| **30.7** | **71** | **13:21:20** | **13:31:50** | **x** | **x** | **x** | | **x** | | **x** | **x** | **x** | **x** | **lidar line** | **4.77** | **292** | **21.27** | **33.9** |
| 30.7 | 72 | 13:39:00 | 13:49:30 | x | x | x | | x | | x | x | x | x | lidar line | 5.44 | 301 | 21.44 | 33.8 |
| 30.7 | 73 | 15:02:30 | 15:13:00 | x | x | x | | x | x | | x | x | x | drone tower | 5.77 | 330 | 21.42 | 34.4 |
| 31.7 | 74 | 07:40:40 | 07:47:00 | x | x | x | | x | x | x | x | x | x | drone line | 4.8 | 317 | 17.15 | 57.3 |
| 31.7 | 75 | 09:20:31 | 09:30:00 | x | x | x | | x | x | x | x | x | x | drone line | 5.02 | 312 | 19.14 | 50.5 |
| 31.7 | 76 | 09:57:30 | 10:07:10 | x | x | x | | x | x | x | x | x | x | drone line | 5.64 | 309 | 19.83 | 48.7 |





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
