# Peer review of "Distributed wind measurements with multiple quadrotor UAVs in the atmospheric boundary layer"

_Atmospheric Measurement Techniques, 2020_

## Author Comment (AC1)

**Distributed wind measurements with multiple quadrotor UAVs in the atmospheric boundary layer**

Tamino Wetz1, Norman Wildmann1, and Frank Beyrich2

1Deutsches Zentrum für Luft- und Raumfahrt e.V., Institut für Physik der Atmosphäre, Oberpfaffenhofen, Germany 2Deutscher Wetterdienst, Meteorologisches Observatorium Lindenberg – Richard-Aßmann-Observatorium, Lindenberg, Germany

Correspondence: Tamino Wetz (tamino.wetz@dlr.de)

**1 Review response**

We want to thank the two anonymous reviewers for their valuable feedback and valid points of criticism to our manuscript.

**1.1 Review Comment 1**

**1.1.1 RC1, General Comments**

In this study, the ability of quadrotors to measure wind speed and wind direction through a relationship with the pitch angle is discussed. While the methodology has already been shown to work well in previous studies, the novel part is the use of multiple quadrotors simultaneously. The authors demonstrate clearly, through comparison with in-situ anemometers along a 99-m tower and a wind lidar, that the use of 'swarms' of drones has a large potential for atmospheric boundary layer studies. The research is presented well, but there are some comments and concerns that need to be addressed.

10

15

20

**1.1.2 RC1, Specific Comments**

1. The word 'swarm' in the context of drones is used a lot throughout the manuscript without defining it. Does it mean two or more drones? Or is there another minimum threshold that he authors define as 'swarm'?

First, again thank you for the positive feedback. We agree on the large potential for atmospheric boundary layer studies.

We agree on that comment and add a short definition at the beginning of the manuscript. We additionally changed the word 'swarm' to 'fleet'. By fleet we mean two to ten drones flying simultaneously. The definition of fleet in this study exclude direct communications in between the drones as the term 'swarm' would imply.

2. line 83: Explain why the flight kinematics of a multirotor with four rotors is more simple than multirotor with more rotors. This may not be obvious to the average reader of this article.

In the kinematic model of the quadrotor only four forces due to the four rotors are acting on the quadrotor. The defined arrangement of the rotors in a square, viewed from the bird perspective, simplifies the model. Multirotors with more than

four rotors have more forces included in the kinematic model. Due to the higher number of rotors, different geometric arrangements of the rotors are necessary which results in a more complex kinematic model. We added this explanation in the revised manuscript.

3. line 85: The argument of choosing a 'racing' drone is not very convincing. There are many settings in the Pixhawk software that one can change/tune to make the multirotor more agile. It is not necessary to select a racing drone in that sense.

We agree on the statement that within the Pixhawk software the controller of the system could be tuned in order to gain agility for the multirotor. The demands on a UAV in our application is to endure strong winds and high turbulence with a good stability of the hover position. In order to sustain the hover position in these conditions, a system with a highly dynamic flight controller and sufficient high actuator performance is required. These requirements are fulfilled with standard settings of commercial racing drones by design. The argumentation is added to the manuscript.

- 4. Page 4: the argument about weight being below the threshold for permits/license requirements, is that still true for the new EU regulations? Or does Germany perhaps still have their own specific regulations?
- Within the new EU regulation of 2019/945 by the EASA (European Union Aviation Safety Agency) currently the "Holybro QAV250" quadrotor can be operated in the "open category" within the subcategory A2 (<2kg) until January 1st,</li>
  2023. After 2023 if the UAV gets a certified id by the manufacture it would be marked as C1 and could be operated in subcategory A1 or if it gets no label it will be listed in the category "privately build" and could be operated in subcategory A3.
  - 5. To make a fair comparison between quadrotors vs other multirotors with more than 4 rotors, one should also include some argument against quadrotors. For example: one may argue that the ability to respond to changes in wind speed is better for more than 4 rotors, and that in case of motor failure of one of the rotors, having more than 4 rotors would be desirable. More arguments can be thought of and should be included.

Choosing a quadrotor before multirotors with more than 4 rotors has several advantages such as easy kinematics, smaller frame sizes and price. Nevertheless, there are disadvantages such as the safety issues in case of motor failure, as flying with three remaining rotors is not possible. Furthermore, the ability to respond to side wind could be smoother and more defined for hexa- or ocotocopters. However, even if strong side winds could not respond imminently by the quadrotor, we will identify the changes in the measured sensor data as copter attitude or accelerometer data. Furthermore, the potential payload is typically higher for multirotors with more than 4 rotors, but since we are not planning to add heavy payloads to the system, we do not consider this relevant for our purpose. Besides the advantages also some disadvantages of quadrotors are now discussed in the revised manuscript.

50

40

45

25

30

6. line 97; include some details of the relevant sensors, including IMU, GPS, HYT271, etc. e.g. accuracy, precision, etc. Best to do that in separate table.

Details about the sensors are identified and listed in a separate table in appendix B of the manuscript. The different

accuracies of the sensors listed in table 1 only represent the raw values of the corresponding data sheets. The filtering and the data fusion of the Pixhawk autopilot is not included in that table. To identify the final accuracy logged by the Pixhawk further studies would be necessary. However, a study about the GPS accuracy during the flight is included in Sect. 4.3.

| sensor       | type                 | accuracy                                                       |
|--------------|----------------------|----------------------------------------------------------------|
| acceleromter | (ICM-20689) / BMI055 | $\pm$ 70 m g                                                   |
| gyroscope    | (ICM-20689) / BMI055 | $\pm 5^{\circ} s^{-1}$                                         |
| magnetometer | IST8310              | $\pm 0.3$                                                      |
| barometer    | MS5611               | $\pm 1.5$ m bar                                                |
| GPS          | ublox NEo-M8N        | 2.5 m (horizontal position) $0.05 \text{ m s}^{-1}$ (velocity) |
| temperature  | HYT271               | $\pm$ 0.2 K (0°C to +60°C)                                     |
| humidity     | HYT271               | $\pm 1.8~\%$ RH at +23°C (0 % RH to 90 % RH)                   |

Table 1. Data sheet of the sensors. Accuracies representing the raw data output of the sensor without any processing of the Pixhawk autopilot.

7. line 104: Did the authors compare wind directions from quadcopter in weathervane mode, vs. quadrotor pointing in one direction and calculating wind direction from pitch and roll (a more common methodology in previous studies)? I would think that in conditions with quite turbulent, varying wind directions, as often encountered in the lower ABL, the weathervane mode may not be optimal.

During the campaign only flights in weathervane mode were performed. However, a prior the campaign we also tested to calculate the wind while the quadrotor was pointing in one direction. In general, the results in weathervane mode looked more promising for calculating the wind speed and direction. Although the weathervane mode is enabled, the roll angle could also be considered for more accurate wind direction calculation in turbulent conditions. We are currently working on a more advanced algorithm, where more sensor data are taken into account for the algorithm, including the roll angle.

- 8. line 107: is there a reference to the software "QGroundControl". Who developed it, how many drones can be controlled by the software? Some more details of this software need to be included.
- Some details are already included in the acknowledgements about the QGroundControl software. QGroundControl is an open-source software developed by the dronecode foundation. The current release of the software allows to control 15 drones simultaneously. However, with minor changes in the source code this number can be increased. This statement and the reference is added to the manuscript.
  - 9. As for air traffic regulations/permits, is one single operator allowed to control more than one quadrotor? If so, is there a limit?
- 75 In the "Annex to Implementing Regulation (EU) 2019/947, PART A UAS OPERATIONS IN THE 'OPEN' CAT-EGORY, AMC1 UAS.OPEN.060(2)(d) Responsibilities of the remote pilot", it is stated that "The remote pilot should

60

55

operate only one UA at a time". Any operation beyond that needs to be applied for with the local authorities and can only be approved on a case-by-case analysis. A risk assessment and procedure of operation needs to be defined that will rule out any harm to people, air traffic and critical infrastructure.

80 10. line 130: Sentence a bit unclear. I think that at least, you have to replace "tips of booms", "to tips of three booms". I understand that at each level, there are three booms and each of these booms has one anemometer. Is that correct? Is that also the case for the wind vanes at the two heights?

Yes, this is fully correct: there are three booms at each level, and each of these booms has one anemometer. At 40m and 98m, each boom also carries a wind vane which is mounted behind the cup. We modified the text to make this clearer.

- 85 11. The sampling rate of anemometers need to be mentioned somewhere. Later on in the paper, I believe it is mentioned that only 1-minute cup data were available which I was a bit surprised about.
   Thanks for pointing at that detail, we added this information to the text.
  - 12. line 137; 76 flights with how many quadrotors each? Give a number or range of numbers here.

The flights were performed with two to ten quadrotors. The range is added to the manuscript and can also be read for each flight in Table C1.

13. line 141: 'hovering for a certain time'. Be more specific. Give at least a time range in minutes.

90

95

100

During the campaign the hovering time was mainly 10 min, but in between 9 min - 11 min. The specific time range is added to the script.

14. line 147: "A safety distance of 20 m to the tower was chosen". That seems quite large, given the possible spatial variability of winds and lack of eddy coherence on those scales. How does that distance translate into a distance from cup anemometer and/or vane? Also, was there a test done to estimate the minimal distance to avoid disturbance of anemometer measurements by the drone?

During the campaign, a special study of the minimal possible distance to the anemometer before disturbance occur at the anemometer data was not performed. However, comparing the mean wind speed of 10 min between UAV and tower measurements, the horizontal distance of 20 meter should not affect the results significantly. Furthermore, for synchronization of the data, it could be thought of a time shift correction in the comparison of the data between anemometers and UAV measurements by calculating a time shift from the mean wind speed and the distance. Nevertheless, the effect of the time shift for 10 min averaged data is small.

15. Page 8, Eq. 9-11: Quite a few variables/parameters are not defined, including p,q,r, and angles phi and theta.

105 The variables p,q,r are defined in Eq. 4 representing the components of angular velocities in the body frame. Furthermore, the angles  $\phi$  and  $\theta$  are defined by Eq. 2 representing the components of the Euler angles in inertia frame. The angle  $\phi$ defines the roll angle and the angle  $\theta$  defines the pitch angle of the Euler angles of the quadrotor. Each single variable is also listed in the nomenclature in the appendix. 16. Section 5: in the general methodology, I am unclear if the pitch/wind speed relationship is based on time-averaged data
or on the higher frequency (1Hz for sonic, 1 minute for cup anemometer) data. This needs to be clarified somewhere, and compared with methodology in previous studies.

For clarification an additional statement is added to manuscript. The calibration for the wind algorithm is based on the time-averaged data of one single flight of approx. 10 min. The idea is to obtain a robust method for calculating the mean wind speed based on the pitch angle of the quadrotor.

115 17. line 231: "Once the offset is determined". It is not clear how the offset is determined. Please explain.

120

125

130

The pitch offset is obtained from the described calibration of the wind speed with the reference tower measurement. The minimization of the wind speed deviation between anemometer and UAV measurements results in the best fit of the pitch offset for the single quadrotors.

18. line 233: Given the large choice of available minimization approaches, can you say a few words about how the solution depends on the specific minimization approach used?

The Trust Region Reflective algorithm was chosen because it allows the definition of bounds in the minimization. The algorithm is well known and stable. From our perspective the choice of the minimization approach will not change the results significantly, concerning the accuracy range of this minimization. Considering the regression function, the minimization approach results in a sufficient solution for the regarded problem. We added a reference to the algorithm in the revised manuscript.

- 19. line 234: "time-averaged wind speed". Since the time average changes from flight to flight and from drone to drone (or is it exactly 10 minutes?), I think there is an issue with a fair comparison here, and the time averaging needs to be normalized somehow. The comparison between cup wind speed and drone wind speed improves with the time averaging (as with other comparisons between independent sensors) and the impact of a varying time-average should be removed in some way. One possibility is to split up all the data in e.g. 1 or 2 minute averages.
- As mentioned before, the time for the hover duration ranges between 9 min and 11 min, but mostly 10 min. However, another comparison with 2 min time averaged data is provided between anemometer and UAV wind measurements. The results plotted in Fig. 1 and Fig. 2 correspond to the shown figures in the manuscript including the same calibration parameter but different evaluation with a time average of 2 min. The corresponding accuracy is listed in table 3 and table 3. As expected the deviation rises with shorter time averages, due to higher uncertainties at smaller scales. The results of 2 min time averaged data are showing comparable trends to the 10 min average evaluation concerning the mean accuracies for different calibration scenarios. The additional evaluation is described in the manuscript. The tables and figures are shown in appendix C.
- 20. Table 2: similar comment to previous one: unless accuracy from each drone is calculated with identical time averaging
  periods, the comparison is not entirely fair. At least some discussion of this issue needs to be included.

The goal of table 2 was to point out that every single drone has a sufficient high accuracy and the system regarded as one

**Table 2.** Accuracy of wind speed measurement in  $[m s^{-1}]$  for a dataset of 34 flights (used for calibration and validation) for a) calibration with all 3 parameters and b) using only pitch offset for calibration with universal parameter values for  $c_p$  and  $c_{d,0}A_0$ . The wind speed accuracy is based on 2 min time averaged data.

|      | individual (Fig. 1) |                   | univ         | ersal             |
|------|---------------------|-------------------|--------------|-------------------|
| #    | $\Delta V_w$        | $\sigma_{ m rms}$ | $\Delta V_w$ | $\sigma_{ m rms}$ |
| 1    | 0.04                | 0.32              | 0.04         | 0.33              |
| 2    | 0.01                | 0.33              | 0.01         | 0.32              |
| 3    | 0.00                | 0.36              | -0.01        | 0.36              |
| 5    | -0.03               | 0.44              | -0.03        | 0.43              |
| 6    | 0.00                | 0.38              | 0.03         | 0.40              |
| 7    | 0.02                | 0.36              | 0.08         | 0.42              |
| 8    | 0.00                | 0.25              | -0.01        | 0.25              |
| 9    | -0.02               | 0.33              | -0.01        | 0.33              |
| 10   | 0.00                | 0.33              | 0.01         | 0.35              |
| mean |                     | 0.34              |              | 0.35              |

**Table 3.** Accuracy of wind speed measurements in  $[m s^{-1}]$  for different calibration data using only pitch offset applied on validation dataset of 12 flights from the second week. The wind speed accuracy is based on 2 min time averaged data.

|      | (a) n34 all (b) n12 fin |                   | first week   | (c) n1 fl.31      |              | (d) n2 fl.31+56 (Fig. 2) |              |                   |
|------|-------------------------|-------------------|--------------|-------------------|--------------|--------------------------|--------------|-------------------|
| #    | $\Delta V_w$            | $\sigma_{ m rms}$ | $\Delta V_w$ | $\sigma_{ m rms}$ | $\Delta V_w$ | $\sigma_{ m rms}$        | $\Delta V_w$ | $\sigma_{ m rms}$ |
| 1    | 0.01                    | 0.28              | 0.01         | 0.28              | -0.20        | 0.35                     | -0.20        | 0.35              |
| 2    | -0.12                   | 0.30              | -0.24        | 0.37              | -0.23        | 0.36                     | -0.01        | 0.28              |
| 3    | -0.13                   | 0.35              | -0.19        | 0.37              | -0.11        | 0.34                     | -0.06        | 0.33              |
| 5    | -0.07                   | 0.45              | -0.22        | 0.50              | -0.31        | 0.55                     | -0.34        | 0.57              |
| 6    | -0.10                   | 0.41              | -0.14        | 0.43              | 0.15         | 0.42                     | 0.15         | 0.42              |
| 7    | 0.10                    | 0.27              | 0.12         | 0.28              | 0.14         | 0.29                     | 0.17         | 0.30              |
| 8    | -0.02                   | 0.28              | -0.02        | 0.28              | -0.03        | 0.28                     | -0.03        | 0.28              |
| 9    | -0.05                   | 0.37              | -0.06        | 0.37              | 0.24         | 0.44                     | 0.23         | 0.43              |
| 10   | 0.02                    | 0.36              | 0.13         | 0.38              | -            | -                        | -0.20        | 0.42              |
| mean |                         | 0.34              |              | 0.36              |              | 0.38                     |              | 0.38              |

Figure 1. 2-minute averaged wind speed for n=34 flights drone vs. tower using the individual parameter calibration from the same 34 flights.

**Figure 2.** 2-minute Average wind-speed for n=12 flights from the second week drone vs. tower using the universal parameter - only individual pitch offset is calibrated from flight number 31 and 56 (scenario (d)).

fleet could provide reliable results for every single drone in the fleet. In general, the uncertainties due to the variability of the atmosphere are at least in the magnitude of the accuracy of the wind measurement, what could be observed by the mean variance of ??? measured by the sonic anemometer during the flight pattern "drone tower". Thus, the difference due to the slightly different time averaging periods is assumed to be small compared to the uncertainties in the atmosphere.

21. line 260-262: can the authors include the analytic relationship they have derived, perhaps in an Appendix? For comparison with past and future studies, this would be very helpful.

The algorithm for calculating the wind speed is already included in the manuscript. Eq. 13 combined with Eq. 15, including the introduced equation for estimating the drag coefficient in Eq. 17 gives the analytic relationship for the calculation of the wind speed.

- 150 of the wind sp
  - 22. line 277: "Flight #31 is selected as the calibration flight as it has an average wind speed of 6 ms-1". Why is 6 m/s average so good to aim for, for a calibration flight? Isn't it more important that the wind speed has both a large average AND a large variance. Otherwise, there may be a potential bias for covering relationship between pitch and the larger wind speeds only.
- We agree on the need of clarification. The sentence evidently did not match the intent of additional information about that flight chosen for the calibration. We want to mention the character of the generic chosen flight for the calculation of the calibration parameters. The idea is, that one can choose a flight with arbitrary wind conditions for the calibration of the pitch offset. We agree that a calibration flight at high wind speeds and large variances could lead to a more precise calibration for high wind speed conditions. However, the goal is to show that it is possible to calibrate the system in frequently occurring wind conditions and still get accurate results in a wide range of different wind conditions.
  - 23. line 291: what typical values or range of values did you encounter for the "offset calibration of the yaw angle" for these drones?

The values for the offset calibration of the yaw angle for the different UAV's range between  $\Delta \psi = -2^{\circ}$  to  $\Delta \psi = 22^{\circ}$  with and average offset angle for all UAV's of  $\Delta \psi = 10.7^{\circ}$ . The information is added to the text.

165 24. Figure 7: Several outliers (at both high and low wind speeds) for the UAV data are clearly visible. Can you discuss? In this specific case the controlling of the Euler pitch angle has two peaks in steering back almost to horizontal copter inclination angle, which results in peaks of small wind speeds. We observe sensor data during that outliers which indicates some movement of the UAV, so that the hover assumption is violated. In future work that is already ongoing we develop more sensitive wind algorithms by taking more sensor data into account to reduce this kind of outliers. In the particular case at 13:40 UTC the sonic anemometer data show high vertical wind up to 3.5 m s-1 that causes lift at the UAV which lead to an increased altitude. In order to sustain the vertical position, the motor thrust is reduced to descent the UAV to the target altitude. To stabilize the descent, the pitch angle is controlled to a more horizontal position. The same situation applies at 13:16 UTC, where UAV measurements also underestimate the reference wind speed.

**2 Relevant changes to the manuscript**

175 We list here the relevant changes to the manuscript:

1. Introduction:

- Text modifications in response to referee comments.

2. Section 2:

- Text modifications in response to referee comments.

180 3. Section 3:

- Text modifications in response to referee comments.

4. Section 4:

- Text modifications in response to referee comments.

5. Section 5:

- Text modifications in response to referee comments.

6. Section 6:

- Text modifications in response to referee comments.

- Figure 7 was modified in response to referee comments.

7. Section 7:

- Text modifications in response to referee comments.

8. Section 8:

- Text modifications in response to referee comments.

9. Appendix:

- Table was added about sensor specification in response to referee comments.

Figures and tables were added for an additional evaluation of the 2 min time averaged data in response to referee comments.

190

---

## Author Comment (AC2)

**Distributed wind measurements with multiple quadrotor UAVs in the atmospheric boundary layer**

Tamino Wetz[1], Norman Wildmann[1], and Frank Beyrich[2]

[1]Deutsches Zentrum für Luft- und Raumfahrt e.V., Institut für Physik der Atmosphäre, Oberpfaffenhofen, Germany
[2]Deutscher Wetterdienst, Meteorologisches Observatorium Lindenberg – Richard-Aßmann-Observatorium, Lindenberg, Germany

**Correspondence:** Tamino Wetz (tamino.wetz@dlr.de)

**1 Review response**

We want to thank the two anonymous reviewers for their valuable feedback and valid points of criticism to our manuscript.

**1.1 Review Comment 2**

**1.2 RC2, General Comments**

5  1. *The paper describes the procedure to use multiple drones for free field measurements. It is well written and includes many details on the drones itself as well as on the methodology for calibrating the drones. Measurements for different flight patterns for the drones are compared to measurements from a met mast with cup and ultrasonic anemometers as well as lidar systems. Even though the analysis is not at its limit, the results are already very promising. Nevertheless I do have a few questions and comments.*

10  Thank you for the positive feedback. Indeed, we are developing our method further and we hopefully produce some more promising results this year.

**1.3 RC2, Specific Comments**

1. *The term „swarm" suggests that the drones are somehow communicating and that one drone-path depends on the path and reaction of the other drone, which is not the case. I would suggest to use another term (unfortunately I do not have*
15  *a better idea).*

For clarification we add a definition of the system to the manuscript. Further, we changed the word from "swarm" to "fleet" of quadrotor. In this particular case a "fleet" only defines two or more drones flying simultaneously without any communication in between the drones.

2. *Would it be possible to use perform the calibration in the a wind tunnel und laminar wind flow? Even though the*
20  *presented calibration function seem to be linear I can imagine that the ambient turbulence and gusts in the wind field*

*might result in an overshoot in the control system which can bias the parameters. In the paper the authors used the 10 minutes averaged data right away — what happens if they use lets say 2 minutes averaged data or 5 minute averaged data for their calibration? Will that increase the error? By looking at shorter time windows the amount of calibration data will automatically increase and maybe will give additional insight in the dynamic response of the drones.*

In general, it should be possible to perform the calibration in wind tunnels, it could even be more accurate concerning the mean wind speed due to the possibility to set a specific wind speed and being independent from changes in the turbulent atmospheric boundary layer. Nevertheless, some adjustments for the hovering mode need to be performed in order to guarantee a specific hover position of the UAV in the wind tunnel, due to GPS failure indoors. The calibration was performed using 10 min averaged data in order to obtain a robust method for calculating the wind speed. Decreasing the time average could increase the accuracy of dynamic behavior. However, besides the advantage of decreasing the average time window, it will also cause new issues concerning the synchronicity of the tower and the UAV data, for example how to handle the time delay of the wind measurements due to the horizontal distance between the compared measurement systems. At least a new evaluation of 2 min time averaged wind data using the parameter of the calibration with 10 min averaged data is provided in appendix C of the manuscript.

3. *The authors mention that they are interested in capturing small-scale structures with an array of drones, which is a very nice idea. Doing that, they should say something about the smallest scale they can resolve with the drones, which is about 0.25m for a single drone, but what about the arrangement of multiple drones? What is the minimum spacing between the drones in horizontal and vertical direction so that the drones do not „feel" the effect of the neighbouring drone? I can imagine that this could be an issue in the vertical direction due to the downwash of the drones. Can the authors comment on that?*

A detail study about the minimal possible distance between the drones before they influence their measurements themselves is not performed yet. Nevertheless, during the flight campaign we performed a pattern with approx. 6 m horizontal distance between the UAVs and we could not observe significant changes in the measurements. The minimal possible horizontal spacing before influencing the neighboring drones also strongly depends on the relative wind direction and the considered flight pattern. Obviously, if the drones are distributed perpendicular or parallel to the wind direction the minimal distance before influencing the neighboring drone will change significantly. The minimal vertical distance also depends on the wind conditions. If the flight is performed in strong winds, the downwash of the drone will drift fast downstream of the mean wind direction. Thus, the drone allocate at lower altitude will not be affect by the downwash as it would be influenced in lower wind conditions. In the present study at the flight pattern "drone tower", what has been used for the calibration, the minimal vertical distance between neighboring drones was 10 m and we have not detected any abnormalities in the data. More detailed studies regarding this topic are out of the scope of this study and will be targeted in more specific studies.

4. *figure 7, b) and c): the vertical positions of the cup and the quadrotors should be marked and maybe separated. It is hard to identify e.g. seven time series in figure 7 c) and it is hard to see which measurement represents which range in vertical*

55      *direction.*

The vertical positions of the anemometers and the quadrotors are additionally marked in figure 7 b) and c).

**2 Relevant changes to the manuscript**

We list here the relevant changes to the manuscript:

1. Introduction:

   – Text modifications in response to referee comments.

2. Section 2:

   – Text modifications in response to referee comments.

3. Section 3:

   – Text modifications in response to referee comments.

4. Section 4:

   – Text modifications in response to referee comments.

5. Section 5:

   – Text modifications in response to referee comments.

6. Section 6:

   – Text modifications in response to referee comments.

   – Figure 7 was modified in response to referee comments.

7. Section 7:

   – Text modifications in response to referee comments.

8. Section 8:

   – Text modifications in response to referee comments.

9. Appendix:

   – Table was added about sensor specification in response to referee comments.

   – Figures and tables were added for an additional evaluation of the 2 min time averaged data in response to referee comments.